# Protective Effect of Low 2-O, 3-O Desulfated Heparin (ODSH) Against LPS-Induced Acute Lung Injury in Mice

**DOI:** 10.3390/biom15091232

**Published:** 2025-08-26

**Authors:** Joyce Gonzales, Rahul S. Patil, Thomas P. Kennedy, Nagavedi S. Umapathy, Rudolf Lucas, Alexander D. Verin

**Affiliations:** 1Division of Pulmonary, Critical Care, and Sleep Medicine, Medical College of Georgia, Augusta University, Augusta, GA 30912, USA; 2Vascular Biology Center, Medical College of Georgia, Augusta University, Augusta, GA 30912, USA; 3Cantex Pharmaceuticals, Inc., Weston, FL 33326, USA; 4GlycoMira Therapeutics, Salt Lake City, UT 84108, USA; 5Department of Medicine, Division of Pulmonary Medicine, Wake Forest University, Winston-Salem, NC 27157, USA; 6Department of Physiology, School of Medicine, Medical College of Georgia, Augusta University, Augusta, GA 30912, USA; 7Department of Pharmacology, School of Medicine, Medical College of Georgia, Augusta University, Augusta, GA 30912, USA

**Keywords:** ODSH, lipopolysaccharide, acute lung injury, acute respiratory distress syndrome, cytokines, lung inflammation

## Abstract

Background: Acute lung injury (ALI) and its severe form, acute respiratory distress syndrome (ARDS), are critical conditions lacking effective pharmacologic therapies. Lipopolysaccharide (LPS), a bacterial endotoxin, is a well-established trigger of ALI. Emerging evidence suggests that heparin derivatives may attenuate lung injury, but their mechanisms remain unclear. Methods: This study evaluated the protective effects of 2-O, 3-O desulfated heparin (ODSH) in a murine model of LPS-induced ALI. Mice received LPS intratracheally with or without ODSH pre-treatment. Lung injury was assessed by bronchoalveolar lavage fluid (BALF) analysis, Evans blue dye albumin EBDA) extravasation, and histopathology. Results: ODSH treatment significantly reduced BALF protein concentration, inflammatory cell infiltration, and EBDA leakage. ODSH preserved endothelial barrier function in vitro, as evidenced by transendothelial electrical resistance (TER) measurements in human lung microvascular endothelial cell (HLMVEC) monolayers. Histological assessment (H&E staining) and myeloperoxidase (MPO) staining demonstrated reduced lung injury and neutrophil infiltration in the ODSH group. ODSH also downregulated pro-inflammatory mediators (NF-κB, IL-6, p38 MAPK) and upregulated the anti-inflammatory cytokine IL-10. Conclusions: ODSH mitigates LPS-induced ALI by reducing vascular permeability, neutrophilic inflammation, and pro-inflammatory signaling while enhancing IL-10 expression. These findings suggest ODSH may offer a novel therapeutic approach for treating ALI.

## 1. Introduction

Acute lung injury (ALI) and the acute respiratory distress syndrome (ARDS) can occur in critically ill patients and are characterized by widespread inflammation and pulmonary permeability edema. Patients become hypoxic and require mechanical ventilation for breathing support. ARDS has many possible causes, both of direct and indirect origin, including severe pneumonia and sepsis, respectively. Although more people are surviving ARDS now than in the past due to improved supportive management and care, mortality remains high at 30–50% [1]. Treatment has been non-pharmacologic with measures recommended by the ARDSNET trials, including low mechanical ventilation tidal volumes, high positive end-expiratory pressure (PEEP), prone positioning and fluid management [2]. Pharmacologic treatment for ARDS remains elusive.

Gram-negative (G-) bacteria, including *Pseudomonas aeruginosa*, *Haemophilus influenzae*, and *Klebsiella pneumoniae* represent common etiological agents of pneumonia in the intensive care units, often leading to ARDS. The G- bacteria contain lipopolysaccharide (LPS), a potent endotoxin comprising part of the outer membrane that is secreted and released upon destruction of the bacterial cell wall. LPS induces an intense inflammatory response in animal and human immune systems, subsequent to pneumonia or sepsis, leading to ARDS [3]. In pneumonia or other insults with G- bacteria in the lungs, the CD14 receptor in monocytes [4] binds LPS. LPS binding protein (LBP), a glycoprotein primarily produced in the liver and lung during the acute phase response, significantly amplifies this interaction by approximately 100 to 1000 fold [5]. Published studies have shown that heparin and unfractionated heparin can bind to LBP and, as such, enhance LPS signal transduction, leading to increased generation of pro-inflammatory cytokines like tumor necrosis factor (TNF) and pro-inflammatory chemokines like Interleukin 8 (IL-8) [4]. The effects of heparin in this study were strictly sulfate-dependent, as desulfated heparins do not bind to LBP nor enhance TNF or IL-8 generation [4]. One such desulfated heparin is 2-O, 3-O desulfated heparin (ODSH), characterized by the removal of sulfate groups at the 2-O and 3-O positions, while retaining the anti-inflammatory properties of native heparin. Following administration, ODSH disrupts the interaction between the receptor for advanced glycation end-products (RAGE) and its ligands, and inhibits key enzymes including heparanase, cathepsin G, and human leukocyte elastase [6,7]. RAGE, a member of the immunoglobulin superfamily, serves as a pivotal mediator of inflammatory responses. In addition to modulating RAGE–ligand interactions, ODSH exhibits inhibitory activity against selectins, thereby impairing tumor cell adhesion to endothelial surfaces and platelets [8]. Unlike native heparin, ODSH does not induce heparin-induced thrombocytopenia (HIT) [7]. Data from the literature have demonstrated that ODSH is protective against lung injury in cystic fibrosis [9]. We have previously shown that it mitigates thrombin-induced endothelial permeability in cell culture [10]. However, the role of ODSH in LPS-induced acute lung injury in mice has never been investigated.

In this study, we evaluated whether the low anticoagulant ODSH can reduce LPS-induced lung vascular leak and inflammation in a well-established murine ALI model. We demonstrate that ODSH pre-treatment mitigates LPS-induced lung injury in mice. The protective ODSH effects correlate with increased production of the anti-inflammatory cytokine IL-10 and with decreased generation of pro-inflammatory cytokines and chemokines (GM-CSF, KC, IL-6, and IL-1α) after LPS insult. We moreover show that ODSH restores the levels of IL-9, IL-15, and IL-17 after LPS treatment, all of which were suggested to participate in immune defense and in the maintenance of lung homeostasis. In correlation with these results, ODSH attenuates upregulation of LPS-induced pro-inflammatory signaling pathways (p38 expression and NF-κB phosphorylation). In summary, our data suggested that ODSH protective effects against LPS-induced ALI in mice are at least partially attributed to the upregulation of anti-inflammatory IL-10 with concomitant suppression of pro-inflammatory cytokine-mediated signaling. These data may pave the way to the development of clinically relevant novel therapeutic candidates for acute lung injury.

## 2. Materials and Methods

### 2.1. Chemicals and Reagents

Unless specified otherwise, all chemicals were sourced from Sigma-Aldrich (St. Louis, MO, USA). LPS derived from *Escherichia coli* serotype O111:B4 was used in this study. The Hanks’ Balanced Salt Solution (HBSS) and phosphate-buffered saline (PBS) were obtained from GIBCO (Invitrogen, Carlsbad, CA, USA). Ammonium-Chloride-Potassium (ACK) Lysing Buffer is from Lonza Bioscience (Allandale, NJ, USA). Myeloperoxidase was obtained from Dako (Carpinteria, CA, USA). Total protein concentrations were determined using the BCA Protein Assay Kit (Pierce Chemical, Rockford, IL, USA), following the manufacturer’s instructions. ODSH was provided in vials of 50 mg/mL (Cantex Pharmaceuticals, Weston, FL, USA).

### 2.2. Transendothelial Electrical Resistance Measurement

Human Lung Microvascular Endothelial Cells (HLMVECs) were obtained from Lonza Bioscience (Allandale, NJ, USA). Transendothelial electrical resistance (TER) was monitored in real time by using electric cell-substrate impedance sensing system (ECIS-zeta, Applied Biophysics, Troy, NY, USA). To measure TER, HLMVEC monolayers, 6 × 10^4^ cells per well, were seeded on 8W10E ECIS arrays and grown to confluency in complete medium (EBM-2/Lonza). The confluent HLMVEC monolayers were treated with either a vehicle or 50 µg/mL ODSH for 40 min, then exposed to 10 eu/mL LPS, and TER was monitored for additional 13 h. The data were adjusted relative to initial resistance measurements and subsequently plotted as normalized resistance values.

### 2.3. Animal Experiments

All experimental procedures adhered to the Animal Welfare Act and received approval from the Institutional Animal Care and Use Committee of Augusta University (IACUC protocol is 2011-0392. The latest approval date is 19 January 2024). CD-1 mice (8–12 weeks old, 20–25 g, n = 4) were procured from Charles River Laboratories (Wilmington, MA, USA). Animals were housed in plastic cages with unrestricted access to food and water. Environmental conditions were maintained at room temperature under a controlled 12 h light/dark cycle.

### 2.4. Animal Surgical Procedure

Mice were anesthetized via intraperitoneal injection of ketamine (150 mg/kg body weight) and acepromazine (15 mg/kg), followed by surgical exposure of the trachea through cervical and thoracic incisions. Lipopolysaccharide (LPS; 2 mg/kg in sterile PBS) was administered intratracheally (IT) using a 20-gauge catheter. Compared to aspiration, IT instillation via neck incision offers more consistent and controlled delivery of LPS directly into the lower airways, minimizing variability in dosing and localization. This method also reduces the risk of upper airway irritation and ensures reliable induction of pulmonary inflammation, and hence is incorporated. Fifteen minutes prior to LPS instillation, animals received either vehicle or ODSH (50 mg/kg) via the internal jugular vein (IJV). Here, we chose the IJV route to ensure rapid systemic delivery and consistent therapeutic availability. In contrast, intraperitoneal (IP) injection of ODSH may result in delayed absorption and variable bioavailability. After 22 h of recovery period, anesthesia was re-induced and Evans Blue Dye Albumin (EBDA; 30 mg/kg in 4% bovine serum albumin) was delivered through the IJV. Following a 120 min circulation period, mice were exsanguinated, the thoracic cavity opened, and the lungs flushed with saline/EDTA via the right ventricle. Lung tissue was harvested for downstream analysis. Bronchoalveolar lavage (BAL) was performed by instilling 1 mL of 10% HBSS; resulting lavage fluid (BALF) and lungs were collected for protein quantification and cell counting. Lung tissues were stored at −80 °C. For histological analysis, right lungs were fixed in 4% paraformaldehyde.

### 2.5. Lung Histology

Lungs were perfused with EDTA to remove blood, then immersed in 4% buffered paraformaldehyde for 18 h to fix the tissue prior to histological assessment using hematoxylin and eosin (H&E) staining. For uniformity, right lung lobes were selected. Tissue sections underwent deparaffinization and rehydration prior to staining. Slides were incubated in Harris Hematoxylin for 15 min, followed by Eosin for 30 s. Subsequently, slides were dehydrated, cleared, and mounted using Cytoseal for microscopic evaluation.

### 2.6. Immunohistochemistry: Myeloperoxidase (MPO)

Lung sections of 5 μm thickness were prepared from paraffin-embedded blocks and mounted on pre-treated slides (Superfrost Plus; VWR Scientific Products, Suwanee, GA, USA). Slides were air-dried overnight and then incubated at 60 °C for 30 min. Deparaffinization was performed using two changes of xylene (7 min each), followed by a graded ethanol series: two changes of absolute ethanol (2 min each), two changes of 95% ethanol (2 min), 80% ethanol (2 min), and 70% ethanol (2 min), transitioning finally to distilled water. Antigen retrieval was carried out using Target Retrieval Solution (pH 6.0; Dako Corp, Carpinteria, CA, USA) in a steamer (Black and Decker rice steamer), followed by rinsing in distilled water. Endogenous peroxidase activity was quenched with 0.3% hydrogen peroxide in distilled water for 5 min, then rinsed with distilled water (2 min) and equilibrated in 1 × PBS (5 min). Slides were incubated with primary antibody against myeloperoxidase (1:2000 dilution) for 30 min at room temperature, followed by two rinses in 1 × PBS. Subsequently, sections were treated with a secondary antibody, peroxidase-conjugated Affinipure F(ab’)_2_ Fragment Donkey anti-rabbit IgG (Jackson ImmunoResearch Laboratories, West Grove, PA, USA), for 1 h at room temperature and rinsed twice in 1 × PBS. Detection was achieved using the DAB Substrate Kit for peroxidase-HRP (Dako Corp, Carpinteria, CA, USA). Counterstaining was performed with hematoxylin (Richard-Allan Scientific, Kalamazoo, MI, USA), and slides were dehydrated, cleared, and prepared for microscopic evaluation.

### 2.7. Lung Permeability Measurement Using Evans Blue Dye Albumin

To evaluate vascular permeability, Evans Blue Dye Albumin (EBDA; 30 mg/kg) was administered via the right internal jugular vein (IJV) two hours prior to study completion. The blood-free left lung was frozen at −20 °C and weighed. EBDA was extracted by homogenizing the tissue in formamide and incubating for 18 h at 60 °C. Following extraction, samples were centrifuged at 5000× *g* for 30 min, and the optical density of the supernatant was measured at wavelengths of 620 nm and 750 nm. EBDA concentrations were derived from a standard calibration curve and normalized to lung tissue weight.

### 2.8. Protein Estimation and Cell Count from the BALF

Bronchoalveolar lavage fluid (BALF) was initially centrifuged at 500× *g* for 15 min at 4 °C. The resulting supernatant underwent a secondary centrifugation at 2500× *g* for 10 min at 4 °C, after which the clarified BALF was used for total protein quantification. Cell pellets were resuspended in ACK Lysing Buffer and incubated for 15 min with gentle agitation, followed by centrifugation at 500× *g* for 15 min at 4 °C. The resulting cell pellet was fixed using 3.7% formalin and subsequently counted using a hemocytometer.

### 2.9. Quantitative Real-Time PCR Analysis (RT-qPCR)

Total RNA was isolated from mouse lung tissue homogenates using TRIzol reagent (Invitrogen, Carlsbad, CA, USA) following standard procedures recommended by the manufacturer. The concentration and purity of the extracted RNA were determined using a NanoDrop spectrophotometer. Subsequently, 2 µg of total RNA was converted to complementary DNA (cDNA) utilizing the High-Capacity cDNA Reverse Transcription kit (Applied Biosystems). Each RT-qPCR assay was conducted in a 20 µL reaction volume using a MicroAmp™ Fast Optical 96-well plate (Applied Biosystems). The reaction setup included 5 µL of cDNA, 10 µL of PowerUp™ SYBR™ Green Master Mix (2×), 3 µL of Nuclease-Free Water, and 2 µL gene-specific primers (1 µL Forward + 1 µL Reverse at 10 µM each final concentration). Amplification was performed using the StepOnePlus™ Real-Time PCR System (Applied Biosystems, Foster City, CA, USA). Gene expression levels were assessed for p38, NF-κB, IL-6, and the reference gene GAPDH. Quantification was achieved using the ΔΔCt method, and relative expression changes were determined using the formula: Relative expression = 2^−ΔΔCt^. Primer sequences are listed in Appendix A.

### 2.10. Protein Extraction and Immunoblotting

Following sacrifice, mouse lungs were dissected and homogenized in lysis buffer supplemented with phosphatase and protease inhibitors (Roche, Indianapolis, IN, USA). Protein samples were subjected to Western blot analysis using the following primary antibodies: anti-mouse p38 and phospho-p38 (both polyclonal, 1:1000), phospho-NF-κB (polyclonal, 1:1000), anti-mouse β-actin (monoclonal, 1:1000), and IL-6 (monoclonal)—all sourced from Cell Signaling Technology (Danvers, MA, USA).

### 2.11. Quantification of Cytokines and Chemokines in BALF

Bronchoalveolar lavage fluid (BALF) was initially centrifuged at 500× *g* for 15 min at 4 °C. The resulting supernatant was subjected to a second centrifugation under the same conditions to ensure purity. Cytokine concentrations in the clarified BALF were quantified using the MCYTOMAG-70K multiplex assay (EMD Millipore-Sigma, Burlington, MA, USA) following the manufacturer’s protocol.

### 2.12. Statistical Analysis

Data are expressed as means ± standard error of the mean (SEM) from 3 to 5 independent experiments. For multiple group comparisons, one-way analysis of variance (ANOVA) followed by post hoc tests was performed. Differences between two groups were assessed using Student’s t-test. Statistical significance was defined as *p* ≤ 0.05.

## 3. Results

### 3.1. ODSH Treatment Significantly Attenuates LPS-Induced Capillary Leak in a Murine Model of LPS-Induced Acute Lung Injury

Elevated microvascular permeability within the lung is a hallmark pathological feature of ALI/ARDS. Mice challenged by intratracheal (IT) instillation of LPS develop lung injury over 24 h [11]. Therefore, we measured the important indicators of the lung permeability, the protein concentration, and the inflammatory cells count in BALF, and the Evans Blue Dye Albumin (EBDA) leak into the lungs. Mice were pretreated i.v. with ODSH via the IJ vein 15 min prior to IT instillation of vehicle or LPS. ODSH treatment significantly reduced protein concentration (Figure 1A) and total cell count (Figure 1B) in BALF. In addition, ODSH significantly reduces LPS-induced extravasation of EBDA into lung parenchyma (Figure 1C).

### 3.2. ODSH Mitigates LPS-Induced Endothelial Permeability in HLMVECs

To examine whether ODSH also affects barrier function of confluent HLMVEC monolayers in vitro, we exposed the cells in ECIS chambers to ODSH for 40 min prior to the addition of LPS and we monitored transendothelial resistance (TER) (an inverse index of endothelial permeability) for 15 h. Figure 2 demonstrated that ODSH increases basal TER and diminishes hyperpermeability induced by LPS, indicating that ODSH strengthens EC barrier function even in the presence of LPS, consistent with the observed protective effect of ODSH in LPS-induced capillary leak in mice (Figure 1).

### 3.3. Histological Evaluation of ODSH Effect on LPS-Induced Lung Inflammation

To substantiate the protective role of ODSH treatment in LPS-induced lung injury, we have performed a histological assessment using hematoxylin and eosin (H&E) staining [11,12]. Figure 3 shows significantly decreased sequestration of neutrophils and diminished injury in murine lung specimens in the LPS/ODSH, control, or ODSH alone groups, as compared to lungs from the LPS group.

### 3.4. Immunohistochemical Analysis of ODSH Effect on LPS-Induced Neutrophil Activation

Myeloperoxidase, an enzyme abundantly expressed in neutrophil granulocytes, serves as a biomarker of neutrophil activity. Elevated MPO levels in bronchoalveolar lavage fluid indicate enhanced infiltration and activation of neutrophils within the pulmonary compartment [3]. MPO was evaluated upon immunohistochemical criteria. The abundance of MPO significantly decreased in LPS/ODSH-treated mice lungs, compared to the LPS-stimulated group (Figure 4). Control or ODSH alone treatment shows minimal MPO staining. Overall, data in Figure 3 and Figure 4 demonstrate that ODSH attenuates LPS-induced lung injury in mice, at least partially by reducing neutrophil activation and infiltration and by strengthening capillary barrier function.

### 3.5. ODSH Suppresses the Activation of Pro-Inflammatory Signaling Induced by LPS in Murine Lung Tissue

To further characterize the effect of ODSH on LPS-induced ALI in mice, we next evaluated the effect of ODSH on inflammation induced by LPS in lung tissue. Mice were challenged with either IT LPS or vehicle 15 min after IJV ODSH, and the level of inflammatory responses (IL-6 and NF-κB /p-NF-κB expressions) was subsequently evaluated by RT-qPCR and western immunoblotting of lung tissue lysates. Figure 5A,B demonstrated that LPS significantly upregulates NF-κB and IL-6 expressions compared to Control and ODSH. These effects were notably attenuated in the LPS/ODSH group, suggesting ODSH’s modulatory impact on inflammatory signaling. Accordingly, Figure 6 demonstrates that mice challenged with LPS alone show increased levels of p-NF-κB and IL-6, both of which are significantly reduced following ODSH pretreatment. Treatment with ODSH alone does not have any effect.

Activation of p38 mitogen-activated protein kinase (MAPK) is a key signaling step in inflammation and is involved in microvascular EC barrier dysfunction [13] and lung inflammatory responses in mice [14]. Figure 5C demonstrated that LPS induces p38 mRNA production, which was significantly reduced upon ODSH treatment, indicating inhibition of p38 MAPK expression by ODSH. To further examine whether ODSH attenuates this important signaling pathway in LPS-induced lung injury in the murine ALI model, the expression of total and activated (phospho form) p38 MAPK was assessed using Western blot analysis with the total and phospho-specific p38 MAPK antibody (Thr180/Thr182), respectively. Figure 7 demonstrates that the expression level of both total and phosphorylated p38 MAPK is significantly elevated in the lung lysates from LPS-stimulated lungs. Addition of ODSH attenuates this effect. While the phospho-p38 and total p38 balance is only modestly (but significantly) affected by ODSH, we believe that, overall, these findings suggest that ODSH may attenuate ALI in mice, at least in part, via attenuation of LPS-driven expression and activation of pro-inflammatory pathways involved in NF-κB and p38 signaling axes.

### 3.6. Effect of LPS and ODSH on the Levels of Cytokines in BALF

To examine whether the protective effect of ODSH on lung injury in the LPS-treated mice occurs due to an inhibitory effect on pro-inflammatory cytokine and chemokine production and/or by stimulation of anti-inflammatory cytokines, we quantified their levels in the BALF using the multiplex MCYTOMAG-70K assay (Millipore-Sigma). Figure 8 demonstrates that while ODSH alone has no significant effect on cytokines in BALF, IT LPS for 24 h drastically increases the pro-inflammatory mediators GM-CSF, IL-1α, and KC in BALF, and IJV pre-treatment with ODSH (15 min prior to LPS) significantly attenuates their generation. These data are consistent with the anti-inflammatory effects of ODSH on lung tissue (attenuation of LPS-induced IL-6 and phospho NF-κB expressions) (Figure 6). In addition, LPS modestly increases pro-inflammatory IL-12p70 expression, but ODSH has no significant effect. Interestingly, while LPS alone minimally increases anti-inflammatory IL-10 production, ODSH pretreatment increases the IL-10 level ~two-fold compared to LPS alone, suggesting that ODSH potentiates the anti-inflammatory response in LPS-induced ALI. Surprisingly, LPS decreases the level of IL-9, IL-15, and IL-17, all proposed to be involved in lung homeostasis, but pre-treatment with ODSH restores their production.

## 4. Discussion

The most aggressive manifestation of Acute lung injury -ARDS- is distinguished by pulmonary permeability edema and capillary barrier dysfunction. LPS, a structural component of the outer membrane of Gram-negative bacteria, functions as a potent endotoxin that elicits a robust immune response in otherwise healthy animals and which can lead to lung endothelial injury, neutrophilic inflammation, and microvascular thrombi [11]. Heparin has long been known to decrease endothelial permeability [15] and target multiple sites of inflammation [7]. A recent review highlights that heparin’s anti-inflammatory properties stem from its capacity to disrupt multiple stages of leukocyte transmigration across inflamed endothelium, as well as its ability to modulate complement system activation [16]. However, the anti-coagulant side effects leading to bleeding and heparin-induced thrombocytopenia (HIT) have limited its therapeutic ability [17,18]. In contrast to native heparin, the derivative 2-O, 3-O-desulfated heparin (ODSH) exhibits potent anti-inflammatory activity with minimal anticoagulant effects [19]. Notably, ODSH has been shown to attenuate neutrophil-driven lung injury in murine models of *Pseudomonas aeruginosa* infection [20]. In this study, we demonstrate that ODSH attenuates LPS-induced acute lung injury and vascular leak, as evidenced by a decrease in protein content and pro-inflammatory cell count in BALF, with a concomitant decrease in the extravasation of EBDA into the lung tissue in mice pre-treated with ODSH prior to LPS insult (Figure 1). Consistently, ODSH mitigates LPS-induced hyperpermeability in HLMVEC monolayers in vitro (Figure 2). In the current study, no overt adverse effects of ODSH were observed in treated mice, consistent with our previous findings and other reports [9,10,20]. Additionally, for safety concerns, common clinical signs such as behavior, body weight, and general appearance were monitored throughout the experimental period, and no abnormalities were detected.

In ALI, increased permeability of pulmonary capillary endothelial cells leads to fluid leakage into the lung parenchyma, often accompanied by neutrophil extravasation and the development of interstitial edema. Neutrophils play a central role in the pathogenesis of pulmonary edema by releasing cytotoxic mediators that disrupt both endothelial and epithelial barriers, exacerbating tissue injury and inflammation [2,21]. Histological evaluations with H&E staining revealed interstitial edema correlating with increased numbers of neutrophils, and both of these were reduced in the ODSH pre-treated lungs (Figure 3). MPO is a peroxidase enzyme in phagocytic cells that produces a bactericidal effect with a broad spectrum of antimicrobial activity (primarily targets Gram-negative bacteria). However, MPO-derived oxidants play a role in microbial defense; their excessive production can contribute to host tissue injury. Elevated MPO levels are associated with the initiation as well as progression of acute vascular inflammation, and their presence correlates with neutrophil-mediated increases in pulmonary vascular permeability [22,23]. In our histological samples, we noted an increased MPO staining in the specimens treated with LPS only compared to those pre-treated with ODSH (Figure 4). These findings suggest that ODSH exerts its barrier protective effects against LPS-induced ALI in mice, at least in part via inhibition of neutrophil activation [24,25].

LPS induced a severe immune response, activating the NF-κB and cytokines that activated lymphocytes and pro-inflammatory cytokines, causing endothelial dysfunction [26,27]. The inflammatory cytokine IL-6 plays a key role in orchestrating immune and inflammatory responses to infections, including the activation and recruitment of leukocytes to the site of inflammation [28,29]. IL-6 is a critical acute phase response cytokine and is a key molecule in ARDS and in LPS-induced lung injury [30].

p38 MAPK is a central signaling molecule activated by inflammatory stimuli such as IL-6. It transduces stress signals to the cell nucleus, triggering transcriptional programs that mediate cellular responses [31]. The mRNA data showed that ODSH attenuated the LPS impact for the levels of NF-κB, IL-6, and p38 genes (Figure 5). The response to LPS endotoxemia stimulation increased p-p38 and induced a dysregulated inflammatory response resulting in lung injury [32,33]. Interestingly, the literature data indicated that LPS upregulates p38, but not ERK or JNK MAPK signaling in murine lung [34]. However, known the involvement of JNK in LPS-induced inflammation, this pathway warrants further investigation. We demonstrated that ODSH significantly decreased phospho-NF-κB (Figure 6), phosphorylated and total p38 (Figure 7), and reduced pro-inflammatory cytokine IL-6 (Figure 6) in murine lung tissue after LPS instillation, thus further supporting the anti-inflammatory function of ODSH in ALI. These findings demonstrate that the anti-inflammatory properties of heparin are retained in ODSH without the worrisome side effects.

Interestingly, IJV pretreatment with ODSH has a differential effect on cytokine production in BALF after IT LPS. It can be subdivided into several groups. In the first group, ODSH significantly attenuates the LPS-induced expression of pro-inflammatory cytokines, IL-1α, as well as the neutrophil attractant chemokine KC (similar to human IL-8) and GM-CSF (Figure 8). IL-1 produces an auto-toxic inflammatory response in the lungs exposed to both viral and bacterial infectious agents. Human manifestations of infection-induced IL-1 overproduction include fever and inflammation in ARDS [35]. It has recently been reported that the inflammatory chemokine IL-8 is significantly elevated in the serum of ALI patients [36]. Anti-IL-8: IL-8 immune complexes isolated from ALI patient fluids activate neutrophils via signaling pathways that include p38 MAPK and Erk [37]. ARDS patients with the hyper-inflammatory phenotype have significantly increased levels of IL-8 in BALF and blood [38]. GM-CSF functions as a cytokine that stimulates stem cells to produce granulocytes and monocytes. Neutrophil priming and activation of GM-CSF contribute significantly to lung tissue injury and the onset of ALI [39]. GM-CSF enhances TLR-4 expression and promotes LPS-induced inflammatory signaling in microglia [40]. It has had pro-inflammatory functions in autoimmune diseases and is a target for antibody therapy [41].

While the mechanisms involved in the inhibitory effects of heparin and its derivatives, like ODSH, on the expression of pro-inflammatory cytokines are largely unknown, early studies suggested that heparin may decrease their mRNA production in human monocytes after an LPS insult [42]. Heparin inhibitory effects on cytokine expression may involve mRNA destabilization/competition with mRNA-binding proteins [43].

In the second group, consisting of IL-12p70, LPS modestly but significantly increases the production of this cytokine, but ODSH addition has no significant effect on LPS-induced increase (Figure 8). Elevation of IL-12p70 in LPS-induced murine ALI is consistent with the literature data [44]. IL-12 has been noted to be significantly elevated in T cells after stimulation with LPS. It is important in the regulation of T cell responses and plays a major role in innate and adaptive immune responses [45,46]. Increased alveolar expression of IL-12p70 was linked to ALI induced by *Klebsiella pneumoniae* infection in mice [47]. It was shown that heparin binds IL-12 in human and murine lymphocytes. However, only in human cells does it lead to an increase in IL-12 bioactivity [48]. Further, heparin-induced IL-12 bioactivity stimulation positively correlates with sulfation level and depends upon heparin chain length and concentration [48], indicating that the effect of heparin on IL-12 activity is complex and species-dependent. While the effects of heparin or ODSH on IL-12 expression in BALF are not described, our data suggest the differential effects of ODSH on cytokine expression in BALF.

In contrast to the first group, ODSH pre-treatment significantly increased production of IL-10 in BALF of LPS-treated mice (Figure 8). IL-10 is recognized as a key plasma factor that suppresses monocyte-driven pro-inflammatory cytokine production and pro-coagulant activity. It functions as a physiological inhibitor that counteracts many LPS-induced responses [49,50], including immune cell activation [51]. Under hyperglycemic stress, heparin downregulates the expression of pro-inflammatory genes while upregulating anti-inflammatory markers such as IL-10 in murine macrophages [52]. While the mechanisms of heparin- or ODSH-induced IL-10 stimulation are unknown, our finding fits well with the described anti-inflammatory features of heparin and its derivatives [16,53,54] and may present an additional anti-inflammatory signaling pathway initiated by heparin.

Surprisingly, while 24 h of IT LPS significantly decreases the expression of IL-9, IL-15, and IL-17, IJV injection of ODSH restores their basal level in BALF (Figure 8). Data from literature demonstrated notable variations in the expression pattern of various cytokines in BALF, depending upon time of treatment [55]. In particular, IL-9 level in BALF shows a cyclic pattern in LPS-induced lung injury in rats and may depend upon major stages of inflammation and resolution [56]. The role of IL-9 in lung injury is controversial. While IL-9 exacerbates lung injury in COPD mice by increasing inflammatory and oxidative stress [57] and airway inflammation associated with SARS-CoV-2 infection [58], it protects against bleomycin-induced lung injury [59] and has a protective role in sepsis-induced lung injury by reducing macrophage apoptosis and M1 polarization [60].

IL-15 regulates the activation and proliferation of T cells, B cells, and natural killer (NK) cells and can be stimulated by LPS through its Toll-like receptors (TLR) [61]. Its anti-inflammatory properties have been used as a vaccine adjuvant in influenzae to promote the immune system [62] and as a vector-based therapy to control melanoma growth in mice [63]. Although there was one study published that showed a conflicting report that IL-15 was involved in worsening influenzae pneumonia [64], the subsequent influenza vaccine has been successfully utilized in the murine model. IL-15 is critical for optimal NK cell production and INF-γ after LPS [65]. A study of macrophage and monocyte involvement of IL-17 shows a pro-inflammatory role. IL-17 is produced by T17 helper cells and, in this study, had an important role in the progression of ALI/ARDS due to macrophage stimulation [66]. However, numerous reports have indicated that IL-17 and Th17 cells are critical to the lung airways’ immune response against various bacterial and fungal infections. Mice deficient in IL-17 receptor expression are susceptible to infection by various pathogens [67]. These results show a dual role for IL-17 in Gram-negative bacterial lung injury/infection [67]. IL-17 was also shown to contribute to neutrophil recruitment and activity in the lung defense against the infection in intracellular pathogens like *Mycoplasma pneumonia* [68]. Apparently, IL-9, IL-15, and IL-17 may have pleotropic and somewhat conflicting roles in ALI. Based on our results, we speculate that ODSH may exert its protective function in the lungs, at least in part, via restoration of the basal level of these cytokines, thus promoting host defense and ALI resolution.

## 5. Conclusions

Our study showed that ODSH treatment attenuates LPS-induced lung injury by reducing lung vascular leak. In addition, ODSH treatment attenuates the LPS-induced neutrophil infiltration and oxidant tissue damage. Concomitantly, ODSH decreases the levels of pro-inflammatory cytokines and chemokines such as IL-6, IL-1α, KC, and GM-CSF and decreases the upregulation of pro-inflammatory signaling pathways, including NF-κB and p38 MAPK in lung tissue. It followed the same trend for NF-κB, IL-6, and p38 mRNA levels. By contrast, ODSH increases the production of anti-inflammatory IL-10 in the BALF of LPS-treated mice. ODSH, moreover, restores the LPS-suppressed levels of IL-9, IL-15, and IL-17 in BALF, suggesting that their expression may be important for lung recovery from an LPS insult. While the role of heparin as an anti-inflammatory agent, which protects against ALI, is well documented [24,54], the mechanisms involved in heparin-mediated regulation of cytokine induction (anti-inflammatory vs. pro-inflammatory) have not been well studied and are an area of future investigation. The use of heparin as an anti-inflammatory agent has been impeded by the fear of bleeding or heparin-induced thrombocytopenia. ODSH mitigates these side effects and appears to maintain the anti-inflammatory properties in LPS-induced acute lung injury. We have shown a beneficial role for low anticoagulant heparin, ODSH, for LPS-induced lung injury in the murine model.

Though these results support the anti-inflammatory potential of ODSH, we acknowledge the fact that further investigation is needed to delineate its effects on specific immune cell populations, including neutrophils, macrophages, eosinophils, and T cells. Moreover, the precise molecular mechanisms by which ODSH modulates inflammatory signaling, particularly within the TLR4–MyD88–NF-κB pathway, remain to be elucidated. Future studies incorporating more advanced techniques and targeted pathway analyses will be essential to fully characterize the immunomodulatory actions of ODSH and its therapeutic relevance in ALI.

## Figures and Tables

**Figure 1 biomolecules-15-01232-f001:**
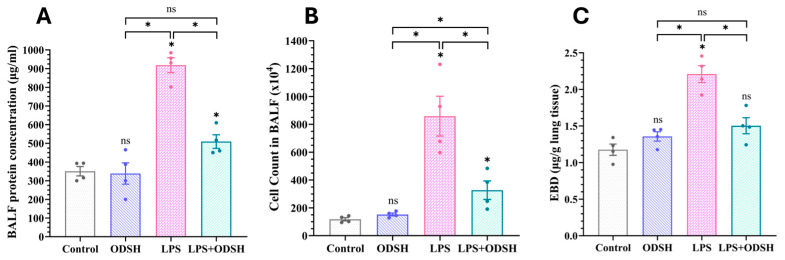
Bronchoalveolar lavage fluid (BALF) protein, cell counts, and Evans Blue Dye Albumin (EBDA) in lung tissue. (**A**) BALF collected at 24 h after treatment was centrifuged, and protein was estimated in the supernatant using Bradford protein estimation kit. ODSH treatment (50 mg/kg given IJV 15 min prior to IT LPS) reduced total protein accumulation in the BALF of LPS (2 mg/kg) induced lung injury. (**B**) ODSH (50 mg/kg given IJV 15 min prior to IT LPS reduces WBC accumulation in the BALF of LPS (2 mg/kg) treated mice compared with untreated mice. BALF was collected at the end of the experiment and centrifuged. Cells were counted using a hemocytometer. ODSH reduced total WBCs in BALF. (**C**) EBDA (30 mg/kg) was injected into IJV 2 h before the termination of the experiment. LPS (2 mg/kg) challenge increased EBDA leakage from the vascular space into surrounding lung tissue in the LPS group, with significant attenuation in the LPS/ODSH group. All values presented as Mean ± SEM (n = 4 mice). ns: not significant, * *p* < 0.05.

**Figure 2 biomolecules-15-01232-f002:**
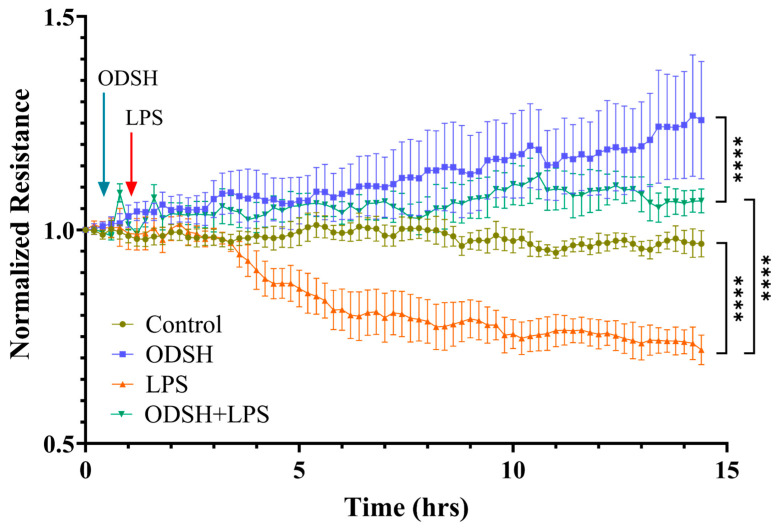
Impact of ODSH on LPS-induced endothelial permeability. ODSH mitigates LPS-induced TER reduction in HLMVEC monolayers. Cells cultured in ECIS arrays were pretreated with 50 µg/mL ODSH for 40 min before exposure to 10 eu/mL LPS. Arrows denote the addition of effectors. TER values were normalized to initial resistance and presented as Mean ± SEM (n = 4). **** *p* < 0.0001.

**Figure 3 biomolecules-15-01232-f003:**
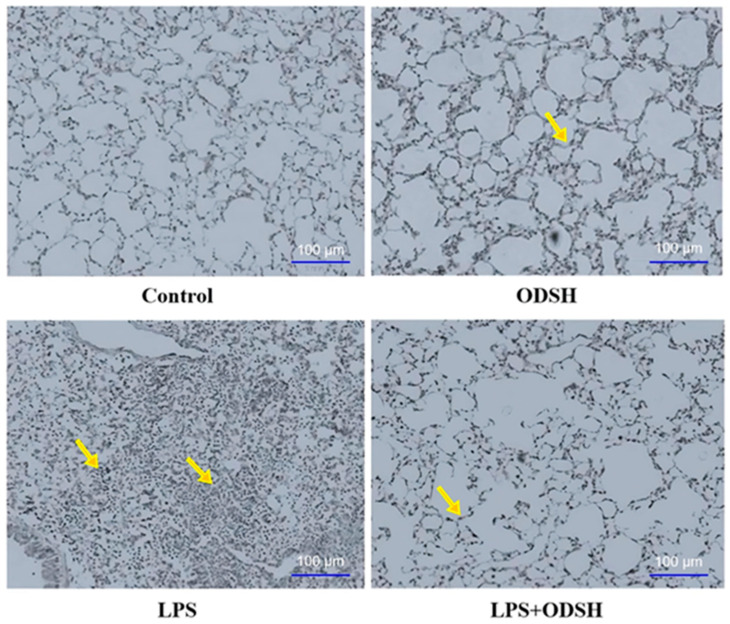
ODSH inhibits LPS-induced inflammatory infiltration in lungs of mice. Representative slides of the lung sections (n = 4 mice) from animal groups of control, ODSH (50 mg/kg), LPS (2 mg/kg), and LPS/ODSH. The pathological changes were examined by hematoxylin and eosin (H&E) staining. Lungs perfused free of blood were immersed in 4% buffered paraformaldehyde at 4 °C for 18 h before H&E staining. Histological analyses of the lung tissue obtained from control mice showed minimal infiltration of neutrophils (scale bar = 100 µm). In contrast, mice exposed to LPS for 24 h produced prominent neutrophil infiltration (arrows) that was attenuated in LPS/ODSH-treated mice. ODSH alone treatment also shows minimal infiltration of the neutrophils.

**Figure 4 biomolecules-15-01232-f004:**
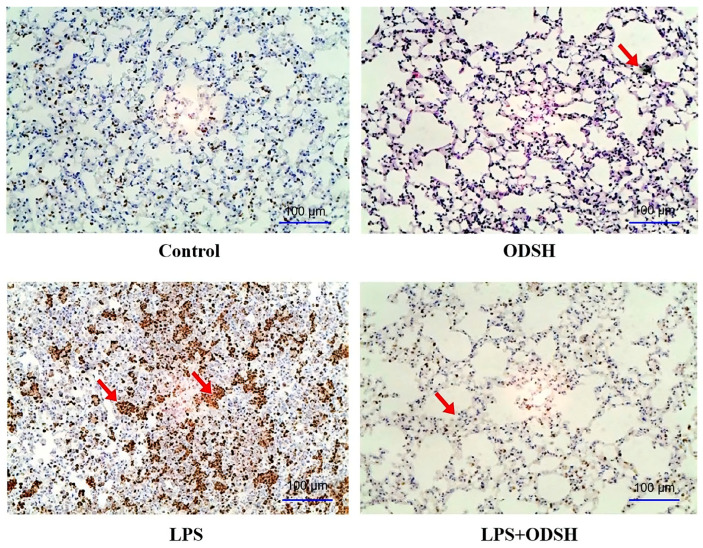
ODSH mediated attenuation of myeloperoxidase (MPO) staining (index of pulmonary infiltration of neutrophils). The MPO staining in the lung tissue (scale bar = 100 µm) indicates a markedly increased infiltration of neutrophils in the lungs of mice from the LPS group which was significantly attenuated in the LPS/ODSH group. The ODSH alone treatment shows a minimal staining comparable to the control (n = 4 mice, representative image from each group).

**Figure 5 biomolecules-15-01232-f005:**
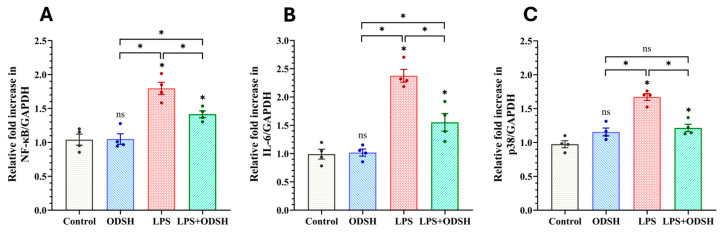
Relative mRNA expression levels of (**A**) NF-κB, (**B**) IL-6, and (**C**) p38 in mouse lung tissue. Expression values were normalized to GAPDH. LPS treatment increased NF-κB, IL-6, and p38 gene expression, which was significantly attenuated in LPS-treated mice administered ODSH, suggesting an inhibitory effect on inflammatory signaling pathways. All values presented as Mean ± SEM (n = 4). ns: not significant, * *p* < 0.05.

**Figure 6 biomolecules-15-01232-f006:**
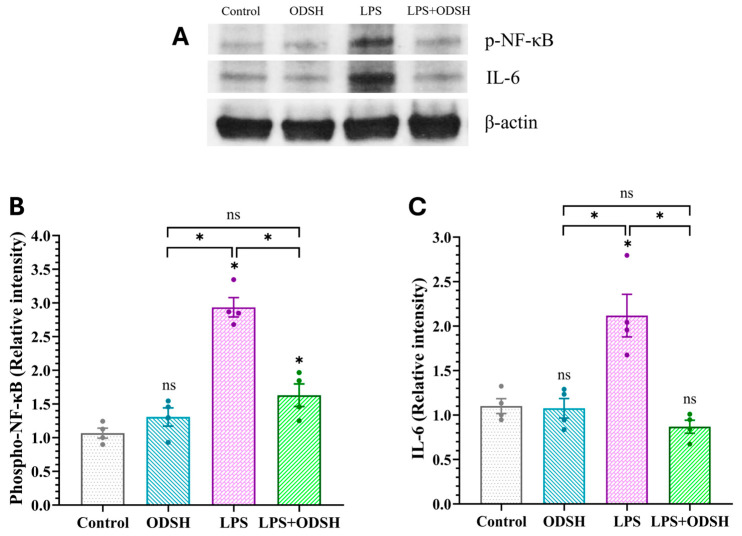
Effects of LPS-induced activation on the expression of NF-κB and IL-6 immunoblot in murine lung lysates. (**A**) Immunoblot protein of p-NF-κB and pro-inflammatory cytokine IL-6 was upregulated in the LPS-only-treated mice and attenuated in mice treated with LPS/ODSH. LPS-induced activation increases expression of phospho-NF-κB and IL-6 compared to control at 24 h. The increase is significantly attenuated by ODSH treatment. See Appendix A for original Western blot images. (**B**,**C**) densitometric representation of Western blot data for p-NF-κB and total IL-6. All values presented as Mean ± SEM (n = 4). ns: not significant, * *p* < 0.05.

**Figure 7 biomolecules-15-01232-f007:**
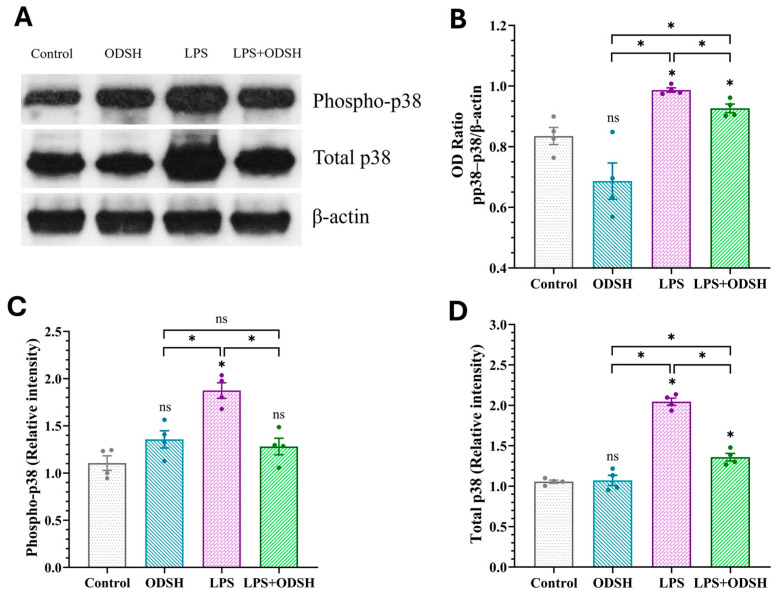
(**A**) Effects of LPS-induced activation on the expression of p-p38 and total p38 immunoblot in murine lung lysates are significantly attenuated by ODSH. Phosphorylated p38, a mitogen-activated protein kinase, responds to stress stimuli from inflammatory cytokines and enhances inflammation. LPS increases expression of p-p38 compared with control 24 h after LPS treatment. The increase is significantly attenuated by ODSH treatment. See Appendix A for original Western blot images. (**B**) Optical density values were normalized to β-actin as the loading control and expressed as the ratio of phospho-p38 to total p38, accounting for total protein expression. LPS significantly increased p38 phosphorylation relative to Control, while treatment with ODSH modestly, but significantly attenuates this effect. (**C**,**D**) Densitometric representation of western blot data for p-p38 and total p38 MAPK. All values presented as Mean ± SEM (n = 4). ns: not significant, * *p* < 0.05.

**Figure 8 biomolecules-15-01232-f008:**
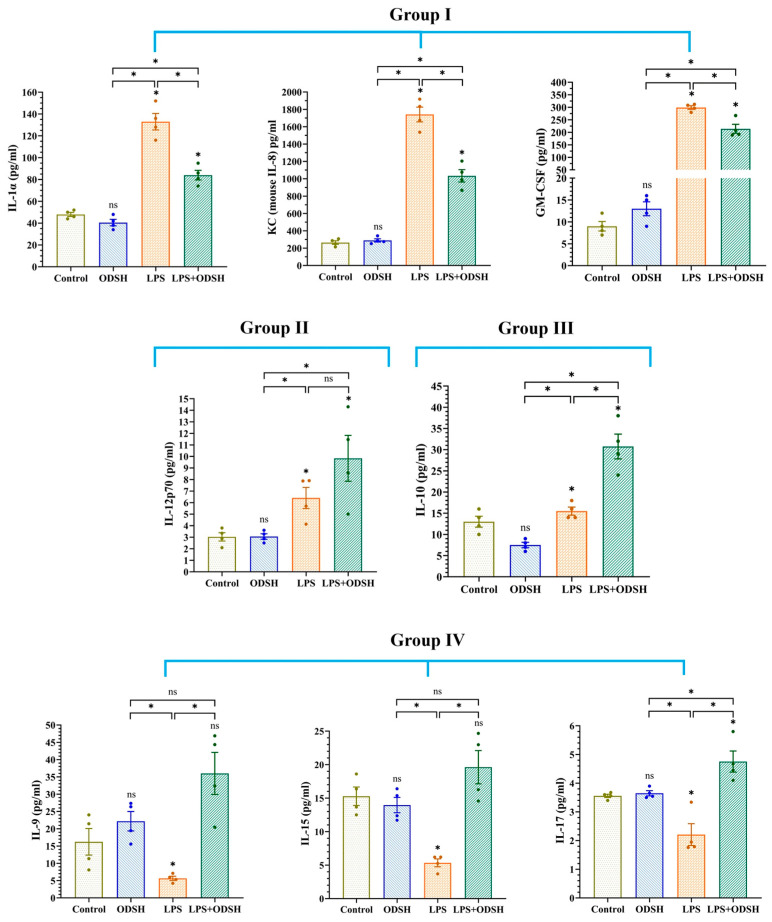
ODSH treatment significantly increases LPS-induced production of anti-inflammatory cytokines and significantly decreases LPS-induced production of pro-inflammatory cytokines in murine bronchoalveolar lavage fluid at 24 h. All values presented as Mean ± SEM (n = 4). ns: not significant, * *p* < 0.05.

## Data Availability

The original contributions presented in the study are included in the article; further inquiries can be directed to the corresponding author.

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
