# Peer review of "Protective Effect of Low 2-O, 3-O Desulfated Heparin (ODSH) Against LPS-Induced Acute Lung Injury in Mice"

_biomolecules, 2025, doi:10.3390/biom15091232_

Round 1

Reviewer 1 Report

Comments and Suggestions for Authors

This is a manuscript by Joyce Gonzales et al. testing the protective effects of 2-O, 3-O de-sulfated heparin (ODSH) in a murine model of LPS-induced ALI. The marker of injury and ODSH protection were assessed by protein leaks, cell count in BALF, and Evans blue dye albumin extravasation, and histopathology. Myeloperoxidase staining neutrophil infiltration also was assessed. Evaluation of ODSH effect on preserving endothelial barrier function was carried out in vitro by trans endothelial electrical resistance measurements in human lung microvascular endothelial cell. They found that ODSH mitigates LPS-induced ALI by reducing vascular permeability, neutrophilic inflammation, and pro-inflammatory signaling while enhancing IL-10 expression. Based on their findings, the authors proposed that ODSH may offer a novel therapeutic approach for treating ALI. This is an interesting topic of research and therapeutic approaches are needed. While this project investigates an interesting topic, it is crucial to address several questions to provide further clarity and benefit the readers.

Comments

  • A brief explanation of the advantages of delivering the LPS through neck incision vs aspiration would be helpful.

  • A statement of the justification of choosing to deliver ODSH through the internal jugular vein (IJV) vs intraperitoneal (IP) route would be helpful to the reader.

  • It is confusing how the effects of ODSH on anti-inflammatory and pro-inflammatory cytokines are subdivided into several groups. What was the basis for choosing this approach?
  • The data should be analyzed with Two-way ANOVA and presented more clearly.

  • Line 115 “CD-1 mice (8-12 weeks old) weighing 20-25 gm” it should be 20- 25 g omit the m.

  • Line 184 “Values are the means ± SE of 3-5 independent experiments” M should be added to ± SE

  • For the western blot, is it 15 µg of protein or 15 µl?

Reviewer 2 Report

Comments and Suggestions for Authors

The manuscript “Protective effect of low 2-O, 3-O desulfated heparin (ODSH)

against LPS-induced acute lung injury in mice” from Joyce Gonzales et al. studies low 2-O, 3-O desulfated heparin (ODSH) against LPS-induced acute lung injury in mice, however, there are several concerns about this manuscript:

  1. In your in vivo studies, n means the number of mice or replicates, please clarify. In Figures 3 and 4, the scale bars are missing. Please add the statistical data. How many replicates did the authors do in Figure 4?
  2. In Figure 1B, ODSH treatment significantly reduced protein concentration and total cell count in BALF. Have you checked other cell types affected (Macrophages, eosinophils, T cells)? Please confirm neutrophil changes during ODSH treatment by flow cytometry and RT-qPCR.
  3. The mechanism by which ODSH attenuates neutrophil infiltration and oxidative tissue injury remains unclear. Given the role of the TLR4-MyD88-NF-κB pathway in LPS-induced inflammation, please investigate whether ODSH interferes directly or indirectly with components of TLR4 signaling. Specifically, evaluate whether ODSH modulates the activity of IκB kinase (IKK) complex, MyD88 adaptor protein, or downstream transcriptional responses.
  4. Did the authors assess lung edema in this model? For example, the wet/dry weight ratio or BALF protein concentration could help evaluate pulmonary fluid accumulation.
  5. Additionally, were any adverse effects of ODSH observed in the treated animals?

Round 2

Reviewer 2 Report

Comments and Suggestions for Authors

The author have addressed my questions.

Author Response

Comments and Suggestions for Authors from reviewer

The author have addressed my questions.

Response: Thank you so much for your evaluation. We have modified the manuscript.